# Effects of the *In ovo* Administration of L-Ascorbic Acid on Tissue L-Ascorbic Acid Concentrations, Systemic Inflammation, and Tracheal Histomorphology of Ross 708 Broilers Subjected to Elevated Levels of Atmospheric Ammonia [†]

Ayoub Mousstaaid [1], Seyed Abolghasem Fatemi [1,*], April Waguespack Levy [2], Joseph L. Purswell [3], Hammed A. Olanrewaju [3], Brittany Baughman [4], Kaylin McNulty [4], Patrick D. Gerard [5] and Edgar David Peebles [1]

1 Department of Poultry Science, Mississippi State University, Mississippi State, MS 39759, USA
2 DSM Nutritional Products, Parsippany, NJ 07054, USA
3 Poultry Research Unit, USDA-ARS, Starkville, MS 39762, USA
4 Department of Pathobiology and Population Medicine, College of Veterinary Medicine, Mississippi State University, Mississippi State, MS 39762, USA
5 Department of Mathematical Sciences, Clemson University, Clemson, SC 29634, USA
* Correspondence: sf1006@msstate.edu
† This publication is a contribution of the Mississippi Agriculture and Forestry Experiment Station. This material is based upon work that is supported by the National Institute of Food and Agriculture, U.S. Department of Agriculture, Hatch project under accession number 1011797. Use of trade names in this publication does not imply endorsement by Mississippi Agricultural and Forestry Experiment Station of these products, nor similar ones not mentioned.

**Abstract:** The effects of *in ovo* injection of L-ascorbic acid (**L-AA**) on tissue L-AA concentrations, systemic inflammation, plasma mineral concentrations, and tracheal histomorphology of Ross 708 broilers subjected to elevated atmospheric ammonia (**NH₃**) levels after hatch were investigated. The four *in ovo* treatments included non-injected (control), saline-injected (control), or saline containing 12 or 25 mg of L-AA. The *in ovo* treatments were applied at 17 days of incubation by injecting a 100 μL volume of each pre-specified treatment into the amnion. At hatch, 12 male chicks were randomly allocated to each of the 12 replicate battery cages belonging to each treatment group. The cages were arranged in a randomized complete block design within a common room. All birds were exposed to 50 ppm of NH₃ at 35 days of posthatch age (**doa**), and the concentration of NH₃ in the room was recorded every 20 s. At 0, 7, 14, 21, and 28 doa, one bird from each cage was arbitrarily selected and euthanized for determinations of liver and eye L-AA concentrations at 0, 7, 14, 21, 28 doa; plasma nitric oxide concentrations at 0, 14, 21, and 28 doa; and plasma calcium and trace mineral concentrations at 0 and 21 doa. Tracheal histomorphology evaluations were performed at 0, 21, and 28 doa. There were no significant treatment differences for plasma nitric oxide and mineral concentrations, and for liver and eye L-AA concentrations at each sampling timepoint. *In ovo* injection of either 12 or 25 mg of L-AA decreased tracheal attenuation incidence at 0 doa compared to the non-injected or saline-injected control groups. Furthermore, the percentage of mild tracheal inflammation scores was lower at 28 doa in response to the *in ovo* injection of 12 mg of L-AA compared to the non-injected or saline-injected control groups. These results indicate that *in ovo* injection of 12 mg of L-AA reduces tracheal inflammation in broilers subjected to elevated atmospheric NH₃.

**Keywords:** inflammation; *in ovo* injection; L-ascorbic acid; ammonia; tracheal histomorphology

## 1. Introduction

*In ovo* injection of broiler hatching eggs has been used commercially to deliver vaccines between 17.50 and 19.25 days of incubation (**doi**) [1]. Previous research has shown that the

amnion or embryo body proper are optimal sites for *in ovo* vaccination [1–3]. Compared to the traditional method of vaccinating live broilers, *in ovo* injection provides a more uniform delivery of vaccine, reduces the risk of contamination during injection, is relatively less stressful for bird development, and is less labor intensive [3]. *In ovo* injection is currently used in the U.S. commercial broiler industry for vaccination against Marek's disease. Additionally, numerous studies have investigated the effects of *in ovo* injection of various nutrients, including L-ascorbic acid (**L-AA**) and vitamin $D_3$, on the hatchability [4–6], live performance [6–12], humoral [12–14] and adaptive immunities [13,15,16], inflammatory response [7,10,16], antioxidant capacity [8,12], and small intestine morphology [14–18] of broilers.

L-ascorbic acid is the active form of vitamin C, which exists under physiological conditions as an ascorbate anion [19]. As a reducing agent, L-AA can donate two electrons from the double bond between the second and third carbon atoms to oxidizing species, or oxidants. Therefore, because L-AA releases electrons, it acts as an antioxidant or free radical scavenger [20]. Electrons from the ascorbate anion can also reduce metals, such as copper (**Cu**) and iron, leading to the formation of super oxides and hydrogen peroxide [21]. Dietary L-AA has been shown to improve the metabolism of calcium (**Ca**) and the binding capacity of proteins [22]. L-ascorbic acid is also required for the conversion of vitamin D into its metabolite form, which accommodates Ca regulation and the calcification process [23]. The *in ovo* supplementation of 4 µg of L-AA in broilers has been shown to increase the levels of ash, phosphorus, and Ca in the femur of broilers at 43 days of age (**doa**) [24]. Additionally, chick-embryo ciliated tracheal organ cultures demonstrated resistance to viral infections after supplementation with L-AA [25]. In humans, supplementary L-AA has resulted in a 15-fold increase in resistance to infectious bronchitis and a 50% reduction in the incidence of respiratory symptoms compared to control treatments [26,27]. These results suggest an immunomodulatory role of L-AA during an infection.

Nitrogen is excreted as uric acid (80%), ammonia (**$NH_3$**; 10%), and urea (5%). The conversion to $NH_3$ is accomplished by microbial degradation and several microbial enzymes in manure [28,29]. Uric acid is the major source of $NH_3$ formation in the ceca of poultry [30–32]. Ammonia is alkaline, corrosive, and recognized as the most abundant toxic gas in poultry houses [33]. The production of $NH_3$ is also affected by the ventilation, temperature, moisture, and pH of litter [34]. Numerous studies have reported the effects of $NH_3$ exposure on broiler performance [35–37]. Broilers subjected to various atmospheric $NH_3$ levels have experienced damage to their respiratory system [36,37], which has been characterized by a loss of tracheal cilia and histopathological changes to the tracheal epithelium [35,38]. Oyetunde et al. [39] demonstrated that exposure to 100 ppm of $NH_3$ for 4 weeks resulted in deciliation of the epithelium of the upper portion of the trachea in broilers. Studies by Al-Mashhadani and Beck [40] and Ritz et al. [41] have also shown deciliation and epithelial hyperplasia in the trachea of birds exposed to elevated levels of $NH_3$. Broilers exposed to 50 ppm of $NH_3$ for 1 to 4 weeks experienced a drop in the number of cilia in their trachea and lungs and an increase in various inflammatory factors throughout their bodies [39–41]. Broilers exposed to 25 ppm of $NH_3$ for 6 weeks can experience mucosal stimulation [42]. In addition, exposing broilers to 60 to 100 ppm of aerial $NH_3$ for an approximate 4-week period can cause respiratory inflammatory reactions, which include tracheitis [43] and airsacculitis [44,45]. These findings indicate that long-term exposure to $NH_3$ has negative effects on tracheal histomorphology and causes a severe inflammatory response in the respiratory system of broilers.

The long-term exposure to $NH_3$ has been shown to result in severe ocular abnormalities and infection due to an increase in oxidative reaction and inflammation [46]. Other associated physiological effects include severe ocular abnormalities [46] and irritation of mucous membranes, leading to corneal ulceration and tracheitis [43,47]. Shi et al. [48] also showed that exposing broilers for 6 weeks to aerial $NH_3$ concentrations between 19.5 and 45.5 ppm caused abnormalities in their tracheal immune response. L-ascorbic acid is considered an antioxidant agent, and it is highly involved in the reduction in systemic

inflammation. In addition, our laboratory found that male broilers have higher L-AA concentration as compared to female at hatch [7]. *In ovo* supplementation of L-AA showed promising but inconsistent results in the live performance of broilers exposed to elevated aerial $NH_3$ concentrations [49]. However, the physiological and morphological mechanisms involved in the recovery of broilers from the negative effects of atmospheric $NH_3$ in response to the *in ovo* administration of L-AA are not well understood. Therefore, the objectives of the study reported herein included a determination of the effects of *in ovo* injection of L-AA on the plasma concentrations of nitric oxide (**NO**), Ca, and the trace minerals copper Cu and zinc (**Zn**) in Ross 708 broilers subjected to elevated levels of atmospheric $NH_3$. Other objectives included a determination of tissue L-AA concentrations and tracheal histomorphology of these same birds. Because $NH_3$ is associated with numerous negative impacts on bird performance, normal tracheal histology, and vision, we hypothesized that L-AA supplementation would help alleviate these detrimental effects.

## 2. Materials and Methods

### 2.1. Egg Incubation

Broiler hatching eggs were collected from 35-week-old commercial Ross 708 broiler breeder hens and stored at 18 °C for 24 h. A total of 65 eggs were randomly set in each of the 4 pre-assigned treatment groups on each of the 6 replicate tray levels (1560 total eggs) in a Chick Master single-stage incubator (Chick Master Incubator Company, Medina, OH, USA) set at 37.5 °C dry bulb and 29.0 °C wet bulb temperatures. Treatment group placement was randomized on each level to avoid any positional effects in the incubator. Air temperature and relative humidity were monitored according to the method described by Fatemi et al. [5]. At 12 and 17 doi, all eggs were candled to remove infertile eggs and those that did not contain live embryos.

At 17 doi, using 100 μL solution volumes, each pre-specified treatment was freshly prepared immediately prior to injection in accordance with procedures described by Zhang et al. [4]. The 4 pre-specified treatments included 1) non-injected control; 2) injection of 100 μL of physiological (0.85% NaCl) saline (sham injection control); 3) injection of 100 μL of saline containing 12 mg of L-AA (**L-AA 12**); or 4) injection of 100 μL of saline containing 25 mg of L-AA (**L-AA 25**). Approximately 350 eggs per each pre-specified treatment were injected using a Zoetis Inovoject® m (Zoetis Animal Health, Research Triangle Park, NC, USA) multi-egg injection machine. The injection needle parameters, as described by Zhai et al. [50], allowed for an injection depth of 2.49 cm originating at the large end of the egg to target the amnion [51,52]. The pre-specified concentrations of L-AA were initially dissolved in saline in each injector infusion bag (400 mL total volume) and were prepared according to the method described by Zhang et al. [4]. Our previous laboratory findings revealed that the L-AA used was safe and resulted in a positive effect on hatching chick quality, with no negative impacts on the hatchability of injected live embryonated eggs (**HI**) and hatchling body weight (**BW**) [7]. After *in ovo* injection at 17 doi, all live embryonated eggs were subsequently assigned hatcher basket positions that corresponded to their arrangements in the setter.

### 2.2. Posthatch Experimental Design and Atmospheric Ammonia Exposure

Hatchlings belonging to the same treatment were pooled across replicate units. From the pool of chicks belonging to a common treatment, 12 male chicks were randomly assigned to individual battery cages for each treatment within each of the 12 replicate blocks. All cages were housed in a common room of a grow-out facility. To eliminate the possibility of the release of additional atmospheric $NH_3$ from used litter, birds were placed in suspended battery cages with feces collected in underlying pans that were cleaned daily. Moderate and constant ventilation was available in the room to further simulate an industry-like environment. Throughout the 5-week period, the broilers were exposed to atmospheric $NH_3$ according to procedures described in detail in previous studies [49,53]. The average

$NH_3$ concentration in the battery room approximated the designated level of 50 ppm. The recorded $NH_3$ levels ranged between 42.5 and 49.7 ppm.

### 2.3. Eye and Liver L-Ascorbic Acid Concentrations

Our laboratory previously found that ocular concentrations of L-AA were significantly higher in male compared to female broiler hatchlings [7]. Additionally, in comparison to saline- and non-injected treatment groups, the L-AA 12 treatment increased ocular L-AA concentrations only in male broilers at 14 doa when they were raised under normal conditions. Therefore, only male broilers were used in the current study. At 0 doa, prior to the assignment of male chicks to battery cages, 2 were randomly selected from each pool for the sampling of both eyes and a liver lobe to determine their L-AA concentrations. Furthermore, at 7, 14, 21, and 28 doa, both eyes and a liver lobe of 2 birds from each replicate battery cage were sampled for determination of their L-AA concentrations. The same lobe of the liver was taken from each bird. The liver and eye L-AA concentrations were measured according to the method described by Mousstaaid et al. [6,7].

### 2.4. Plasma NO, Calcium, and Trace Mineral Analyses

For systemic inflammatory response evaluation, plasma NO concentrations were determined at 0, 14, 21, and 28 doa. Moreover, plasma Ca, Cu, and Zn concentrations were determined at 0 and 21 doa. For all analyses, 12 birds per treatment (1 male bird per treatment replicate cage) were randomly selected to be individually weighed and bled by venipuncture of the wing brachial vein. All blood samples were collected into K2 EDTA tubes (Fisher Scientific, Hampton, NH, USA), and plasma was extracted by centrifugation ($3000 \times g$ for 15 min at 4 °C). Approximately 0.5 mL of the plasma was stored at $-20$ °C in plastic microcentrifuge tubes for NO and trace mineral analysis. Plasma NO concentrations were determined using the method described by Bowen et al. [54]. The concentrations of Ca, Cu, and Zn were analyzed using ICP-MS. The analysis included use of 5 mL of trace metal nitric acid in microwave digestion tubes with 2 mL of 30% hydrogen peroxide as a digestion reagent. Mineral analyses were conducted at the Mississippi State University Chemical Laboratory, as previously described by Laur et al. [55].

### 2.5. Histopathologic Examination

At 0, 21, and 28 doa, one bird from each replicate cage in each treatment group was randomly selected and euthanized by carbon dioxide asphyxiation for histopathological assessment. Trachea tissue samples were then collected and placed into 10% neutral buffered formalin. The pathologist conducting the tissue evaluations was unaware of bird treatment origin. The trimmed tracheal specimens were then processed routinely, sectioned at 5 µm, and stained with hematoxylin and eosin. Microscopic tracheal sample thickness measurements were taken from 3 similar locations. Any signs of ulceration, hyperplasia, or metaplasia were avoided to obtain a mean, which represents most of the mucosa for the tracheal measurements. Inflammation consisted of lymphocytic and heterophilic infiltrates [56]. Ulceration of the squamous epithelium is categorized by complete loss of the epithelium and the basement membrane with exposure of the underlying submucosa [57]. The submucosa contains tracheal glands, which secrete mucous to facilitate the movement of particles within the lumen of the trachea [58,59]. In addition, attenuation is defined as a loss in the height of ciliated epithelial cells [59]. The scoring systems of the tracheal samples are listed in Table 1. Lower scores for all these variables are considered improvements in tracheal histomorphology. Mean scores as well as percentages of the different scores for each trachea variable are presented. Tracheal thickness (µm) was recorded as an average of three measurements in different locations within each sample. Micrographs depicting 2 types of trachea cilial erosion, gland assessment, inflammation, attenuation, and ulceration are provided in Figure 1.

**Table 1.** Scoring system used for histopathologic examination of trachea samples.

| | |
|---|---|
| Cilial erosion | 0 = normal, 1 = focal (mild), 2 = multifocal (moderate), 3 = diffuse (severe) |
| Gland assessment | 0 = normal, 1 = depletion (mild), 2 = hyperplasia (moderate), 3 = both depletion and hyperplasia (severe) |
| Inflammation | 0 = normal, 1 = mild lymphocytes, 2 = moderate lymphocytes, 3 = heterophils/moderate lymphocytes |
| Attenuation | 0 = normal, 1 = focal (mild), 2 = multifocal (moderate), 3 = diffuse (severe) |
| Ulceration | 0 = normal, 1 = focal (mild), 2 = multifocal (moderate), 3 = diffuse (severe) |

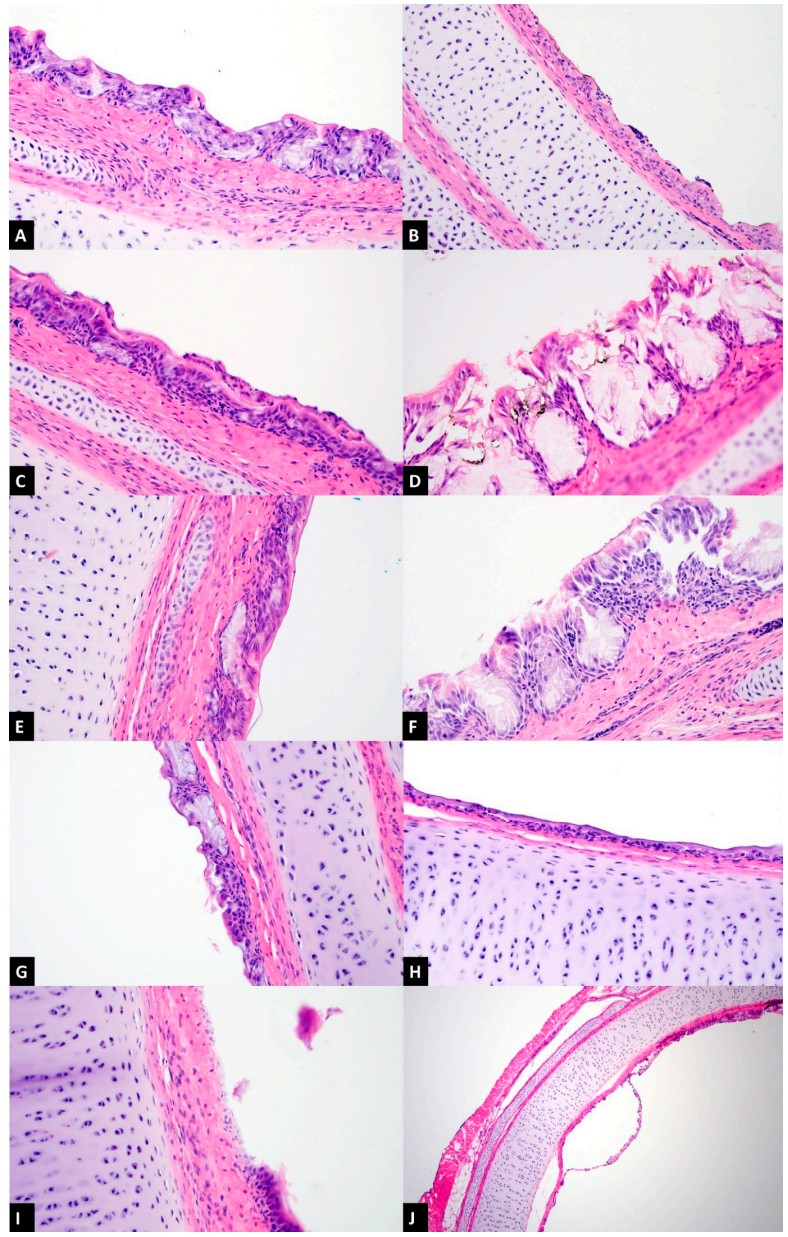

**Figure 1.** (**A**) Mild cilial erosion, 40×, stained using hematoxylin and eosin (H&E). (**B**) Severe cilial erosion, 40×, H&E. (**C**) Gland depletion, 40×, H&E. (**D**) Gland hyperplasia, 40×, H&E. (**E**) Mild inflammation, 40×, H&E. (**F**) Moderate inflammation, 40×, H&E. (**G**) Mild epithelial attenuation, 40×, H&E. (**H**) Severe epithelial attenuation, 40×, H&E. (**I**) Mild epithelial ulceration, 40×, H&E. (**J**) Moderate epithelial ulceration, 10×, H&E.

*2.6. Statistical Analysis*

The experimental design was a randomized complete block. Individual birds served as the unit of treatment replication for the eye and liver L-AA concentrations at 0 doa. Battery cages served as the unit of treatment replication for the eye and liver L-AA concentrations after 0 doa and for all other data at all time periods. A group of battery cages was the blocking factor, with all the *in ovo* injection treatments being randomly represented in each block. All data within each individual period were separately analyzed by ANOVA using the procedure for linear mixed models (PROC GLIMMIX) of SAS 9.4© [60]. The following model was used for analysis of the data:

$$Y_{ij} = \mu + B_i + T_j + E_{ij},$$

where $\mu$ was the population mean; $B_i$ was the replicate or block factor (i = 1 to 12); $T_j$ was the effect of each *in ovo* injection treatment (j = 1 to 4); and $E_{ij}$ was the residual error. Means separations were performed by Fisher's protected least significant difference [61]. Differences were considered significant at $p \leq 0.05$.

**3. Results**

In a companion study where the same eggs were used [50], it was confirmed that the frequencies of the injections across treatments were 93.4% and 6.6% in the amnion and embryo body proper, respectively.

*3.1. Concentrations of Eye and Liver L-AA, and Plasma NO, Ca, Cu, and Zn*

No significant treatment differences for eye and liver L-AA concentrations were observed at 0, 7, 14, 21, and 28 doa (Table 2), or for NO concentrations at 0, 14, 21, and 28 doa (Table 3). Additionally, no significant treatment differences were observed for plasma concentrations of Ca, Cu, and Zn at 0 and 21 doa (Table 4).

**Table 2.** Effects of treatment (non-injected; saline-injected (**saline**); saline containing 12 mg of L-ascorbic acid (**L-AA 12**) or 25 mg of L-ascorbic acid (**L-AA-25**)) on Ross 708 broiler L-AA concentration in the eye and liver from 0 to 28 days of posthatch age (**doa**).

| Treatment | Eye | | | | |
| | 0 doa | 7 doa | 14 doa | 21 doa | 28 doa |
| | ------------------------------- (µmol) ------------------------------- | | | | |
| --- | --- | --- | --- | --- | --- |
| Non-injected [1] | 4.48 | 4.51 | 2.94 | 3.09 | 2.15 |
| Saline [2] | 4.15 | 4.23 | 3.25 | 3.41 | 2.16 |
| L-AA 12 [3] | 3.96 | 4.25 | 3.27 | 3.05 | 1.99 |
| L-AA 25 [4] | 4.47 | 3.82 | 2.95 | 3.17 | 1.97 |
| Pooled SEM | 0.456 | 0.488 | 0.291 | 0.295 | 0.283 |
| *p*-Value | 0.607 | 0.795 | 0.522 | 0.628 | 0.861 |
| | Liver | | | | |
| | 0 doa | 7 doa | 14 doa | 21 doa | 28 doa |
| | ------------------------------- (µmol) ------------------------------- | | | | |
| Non-injected | 24.6 | 20.3 | 14.8 | 12.7 | 7.09 |
| Saline | 24.8 | 20.2 | 14.0 | 12.2 | 8.22 |
| L-AA 12 | 27.1 | 20.9 | 16.9 | 11.9 | 7.53 |
| L-AA 25 | 27.4 | 21.1 | 16.1 | 12.4 | 8.26 |
| Pooled SEM | 1.36 | 2.12 | 3.02 | 1.34 | 0.908 |
| *p*-Value | 0.109 | 0.968 | 0.773 | 0.948 | 0.511 |

[1] Eggs that were not injected. [2] Eggs that were injected with 100 µL saline at 17 doi. [3] Eggs that were injected with 100 µL saline containing L-AA 12 at 17 doi. [4] Eggs that were injected with 100 µL saline containing L-AA 25 at 17 doi. N = Two replicate birds in each treatment were used for means calculations at 0 doa. Two birds in each of the 12 replicate groups in each treatment were used for means calculations at 7, 14, 21, and 28 doa.

**Table 3.** Effects of treatment (non-injected; saline-injected (**saline**); saline containing 12 mg of L-ascorbic acid (**L-AA 12**) or 25 mg of L-ascorbic acid (**L-AA 25**) administered at 17 days of incubation (**doi**)) on Ross 708 broiler plasma nitric oxide (**NO**) concentrations at 0, 14, 21, and 28 days of age (**doa**).

| Treatment | 0 doa | 14 doa | 21 doa | 28 doa |
|---|---|---|---|---|
| | -------------------------------- (μmol) -------------------------------- | | | |
| Non-injected [1] | 7.04 | 6.93 | 6.05 | 5.90 |
| Saline [2] | 6.78 | 6.68 | 6.48 | 6.97 |
| L-AA 12 [3] | 6.95 | 7.52 | 7.19 | 6.17 |
| L-AA 25 [4] | 7.09 | 5.55 | 9.13 | 5.59 |
| Pooled SEM | 1.135 | 0.745 | 0.958 | 0.846 |
| *p*-Value | 0.998 | 0.317 | 0.154 | 0.661 |

[1] Eggs that were not injected. [2] Eggs that were injected with 100 μL saline at 17 doi. [3] Eggs that were injected with 100 μL saline containing L-AA 12 at 17 doi. [4] Eggs that were injected with 100 μL saline containing L-AA 25 at 17 doi. N = One bird in each of the 12 replicate groups in each treatment–doa combination was used for means calculations.

**Table 4.** Effects of treatment (non-injected; saline-injected (**saline**); saline containing 12 mg of L-ascorbic acid (**L-AA 12**) or 25 mg of L-ascorbic acid (**L-AA 25**) administered at 17 days of incubation (**doi**)) on Ross 708 broiler calcium (**Ca**), copper (**Cu**), and zinc (**Zn**) concentrations at 0 and 21 days of age (**doa**).

| Treatment | 0 doa | | |
|---|---|---|---|
| | Ca (ppm) | Cu (ppm) | Zn (ppm) |
| Non-injected [1] | 100 | 0.06 | 3.25 |
| Saline [2] | 102 | 0.07 | 3.39 |
| L-AA 12 [3] | 102 | 0.08 | 3.31 |
| L-AA 25 [4] | 101 | 0.10 | 3.34 |
| Pooled SEM | 2.6 | 0.056 | 0.313 |
| *p*-Value | 0.805 | 0.809 | 0.977 |
| | **21 doa** | | |
| Non-injected | 92 | 0.09 | 2.35 |
| Saline | 97 | 0.06 | 2.37 |
| L-AA 12 | 90 | 0.08 | 2.22 |
| L-AA 25 | 99 | 0.09 | 2.30 |
| Pooled SEM | 6.9 | 0.018 | 0.133 |
| *p*-Value | 0.769 | 0.596 | 0.865 |

[1] Eggs that were not injected. [2] Eggs that were injected with 100 μL saline at 17 doi. [3] Eggs that were injected with 100 μL saline containing L-AA 12 at 17 doi. [4] Eggs that were injected with 100 μL saline containing L-AA 25 at 17 doi. N = One bird in each of the 12 replicate groups in each treatment–doa combination was used for means calculations.

### 3.2. Tracheal Histomorphology

Only those treatment differences that were significant are noted. At 0 doa, the L-AA 12 and 25 mg *in ovo* injected treatment groups had a lower mean score for the level of tracheal attenuation compared with the non-injected and saline-injected controls. Conversely, at 21 doa, the L-AA 12, L-AA 25, and saline-injected groups had a higher mean score for the level of tracheal attenuation in comparison to the non-injected group (Table 5). At 0 doa, score 1 incidence for tracheal attenuation and score 2 incidence for tracheal ulceration were significantly lower for birds in the saline-injected, L-AA 12, and L-AA 25 treatments in comparison to those in the non-injected treatment group (Table 6). At 21 doa, score 1 incidence for tracheal attenuation was observed to be lower in the saline control and L-AA 12 and L-AA 25 treatment groups in comparison to the non-injected control group. Conversely, score 2 incidences for tracheal attenuation were significantly higher in the saline control, L-AA 12, and L-AA 25 treatment groups compared to the non-injected control group (Table 7). At 28 doa, the L-AA 12 *in ovo* supplemented group had a higher

incidence of 0 scores for tracheal glands than both control groups, with the L-AA 25 group being intermediate. Score 1 incidence for inflammation was lower in the L-AA 12 treatment in comparison to the L-AA 25 and non-injected and saline-injected control treatment groups. In addition, the saline control, L-AA 12, and L-AA 25 treatment groups had lower score 3 incidence for tracheal attenuation compared to the non-injected control group (Table 8).

**Table 5.** Effects of treatment (non-injected; saline-injected (**saline**); saline containing 12 mg of L-ascorbic acid (**L-AA 12**) or 25 mg of L-ascorbic acid (**L-AA 25**) administered at 17 days of incubation (**doi**)) on mean score [1] for various tracheal histomorphological variables (cilial erosion, glands, inflammation, attenuation, and ulceration) at 0, 21, and 28 days of posthatch age (**doa**).

| Treatment | Cilial Erosion [6] | Glands [7] | Inflammation [8] | Attenuation [9] | Ulceration [10] |
|---|---|---|---|---|---|
| | | | **0 doa** | | |
| Non-injected [2] | 1.50 | 0.58 | 0.58 | 2.42 [a] | 0.83 |
| Saline [3] | 0.83 | 1.00 | 1.00 | 2.08 [a] | 0.50 |
| L-AA 12 [4] | 1.33 | 1.00 | 1.00 | 1.25 [b] | 0.25 |
| L-AA 25 [5] | 0.83 | 0.83 | 0.83 | 1.25 [b] | 0.17 |
| Pooled SEM | 0.307 | 0.265 | 0.265 | 0.391 | 0.265 |
| *p*-Value | 0.072 | 0.359 | 0.359 | 0.007 | 0.068 |
| | | | **21 doa** | | |
| Non-injected | 0.67 | 1.00 | 1.67 | 0.92 [b] | 0.33 |
| Saline | 0.58 | 0.92 | 0.92 | 1.58 [a] | 0.25 |
| L-AA 12 | 0.58 | 1.33 | 1.33 | 2.08 [a] | 0.17 |
| L-AA 25 | 0.75 | 1.17 | 1.17 | 1.67 [a] | 0.25 |
| Pooled SEM | 0.363 | 0.30 | 0.319 | 0.298 | 0.221 |
| *p*-Value | 0.961 | 0.514 | 0.632 | 0.004 | 0.902 |
| | | | **28 doa** | | |
| Non-injected | 1.25 | 0.92 | 1.33 | 0.92 | 0 |
| Saline | 1.33 | 0.92 | 1.58 | 0.92 | 0 |
| L-AA 12 | 0.58 | 1.1 | 2.08 | 1.08 | 0 |
| L-AA 25 | 1.00 | 1.0 | 1.92 | 1.00 | 0.25 |
| Pooled SEM | 0.368 | 0.220 | 0.310 | 0.220 | 0.127 |
| *p*-Value | 0.187 | 0.851 | 0.086 | 0.851 | 0.137 |

[a,b] Treatment means within the same variable column within type of treatment with no common superscript differ significantly ($p \leq 0.05$). [1] Mean score is the average of scores (0, 1, 2, 3) for each variable. [2] Eggs that were not injected with solution. [3] Eggs that were injected with 100 µL saline at 17 doi. [4] Eggs that were injected with 100 µL saline containing L-AA 12 at 17 doi. [5] Eggs that were injected with 100 µL saline containing L-AA 25 at 17 doi. [6] Cilial erosions were scored as follows: 0 = (normal), 1 = focal (mild), 2 = multifocal (moderate), 3 = diffuse (severe). [7] Tracheal glands were scored as follows: 0 = normal, 1 = depletion (mild), 2 = hyperplasia (moderate), 3 = both depletion and hyperplasia (severe). [8] Tracheal inflammation was scored as follows: 0 = normal, 1 = mild lymphocytes, 2 = moderate lymphocytes, 3 = heterophils/moderate lymphocytes. [9] Attenuations were scored as follows: 0 = normal, 1 = focal (mild), 2 = multifocal (moderate), 3 = diffuse (severe). [10] Ulcerations were scored as follows: 0 = normal, 1 = focal (mild), 2 = multifocal (moderate), 3 = diffuse (severe). N = One bird in each of the 12 replicate groups in each treatment–doa combination was used for means calculations.

**Table 6.** Effects of treatment (non-injected; saline-injected (**saline**); saline containing 12 mg of L-ascorbic acid (**L-AA 12**) or 25 mg of L-ascorbic acid (**L-AA 25**) administered at 17 days of incubation (**doi**)) on percentages of 0, 1, 2, and 3 scores for various tracheal histomorphological variables (cilial erosion, glands, inflammation, attenuation, ulceration, and thickness) at 0 days of posthatch age (**doa**).

| | | Non-Injected [1] | Saline [2] | L-AA 12 [3] | L-AA 25 [4] | Pooled SEM | *p*-Value |
|---|---|---|---|---|---|---|---|
| | | | | **Treatments** | | | |
| Cilial Erosion [5] (%) | Score 0 | 16.7 | 25.0 | 8.3 | 36.4 | 17.13 | 0.261 |
| | Score 1 | 25.0 | 66.7 | 50.0 | 36.4 | 20.10 | 0.180 |
| | Score 2 | 50.0 | 8.0 | 41.7 | 27.3 | 18.69 | 0.130 |
| | Score 3 | 8.3 | 0 | 0 | 0 | 5.89 | 0.402 |

**Table 6.** *Cont.*

| | | Non-Injected [1] | Saline [2] | L-AA 12 [3] | L-AA 25 [4] | Pooled SEM | *p*-Value |
|---|---|---|---|---|---|---|---|
| | | | | **Treatments** | | | |
| Glands [6] (%) | Score 0 | 50.0 | 25.0 | 8.3 | 27.3 | 17.99 | 0.152 |
| | Score 1 | 41.7 | 58.3 | 83.3 | 72.7 | 19.37 | 0.212 |
| | Score 2 | 8.3 | 8.3 | 8.3 | 0 | 10.32 | 0.822 |
| | Score 3 | 0 | 8.3 | 0 | 0 | 5.96 | 0.402 |
| Inflammation [7] (%) | Score 0 | 50.0 | 16.7 | 8.3 | 27.3 | 17.35 | 0.103 |
| | Score 1 | 41.7 | 66.7 | 83.3 | 72.7 | 19.12 | 0.210 |
| | Score 2 | 8.3 | 8.3 | 8.3 | 0 | 10.32 | 0.800 |
| | Score 3 | 0 | 8.3 | 0 | 0 | 5.96 | 0.402 |
| Attenuation [8] (%) | Score 0 | 8.3 | 0.0 | 8.3 | 9.1 | 10.31 | 0.801 |
| | Score 1 | 58.3 [a] | 41.7 [a] | 8.3 [b] | 0 [b] | 16.17 | **0.010** |
| | Score 2 | 25.0 | 58.3 | 50.0 | 72.7 | 20.03 | 0.208 |
| | Score 3 | 8.3 | 0.0 | 33.3 | 18.2 | 14.22 | 0.123 |
| Ulceration [9] (%) | Score 0 | 50.0 | 58.3 | 75.0 | 81.8 | 19.49 | 0.306 |
| | Score 1 | 16.7 | 33.3 | 25.0 | 18.2 | 17.85 | 0.783 |
| | Score 2 | 33.3 [a] | 8.3 [b] | 0 [b] | 0 [b] | 11.79 | **0.019** |
| | Score 3 | 0 | 0 | 0 | 0 | - | - |
| Thickness [10] (µm) | | 25.8 | 24.4 | 25.7 | 20.9 | 2.44 | 0.186 |

[a,b] Treatment means within the same variable column within type of treatment with no common superscript differ significantly ($p \leq 0.05$). [1] Eggs that were not injected with solution. [2] Eggs that were injected with 100 µL saline at 17 doi. [3] Eggs that were injected with 100 µL saline containing L-AA 12 at 17 doi. [4] Eggs that were injected with 100 µL saline containing L-AA 25 at 17 doi. [5] Cilial erosions were scored as follows: 0 = (normal), 1 = focal (mild), 2 = multifocal (moderate), 3 = diffuse (severe). [6] Tracheal glands were scored as follows: 0 = normal, 1 = depletion (mild), 2 = hyperplasia (moderate), 3 = both depletion and hyperplasia (severe). [7] Tracheal inflammation was scored as follows: 0 = normal, 1 = mild lymphocytes, 2 = moderate lymphocytes, 3 = heterophils/moderate lymphocytes. [8] Attenuations were scored as follows: 0 = normal, 1 = focal (mild), 2 = multifocal (moderate), 3 = diffuse (severe). [9] Ulcerations were scored as follows: 0 = normal, 1 = focal (mild), 2 = multifocal (moderate), 3 = diffuse (severe). [10] Tracheal thickness (µm) was recorded as an average of three measurements in different locations within each sample. N = One bird in each of the 12 replicate groups in each treatment–score combination was used for means calculations.

**Table 7.** Effects of treatment (non-injected; saline-injected (**saline**); saline containing 12 mg of L-ascorbic acid (**L-AA 12**) or 25 mg of L-ascorbic acid (**L-AA 25**) administered at 17 days of incubation (**doi**)) on percentages of 0, 1, 2, and 3 scores for various tracheal histomorphological variables (cilial erosion, glands, inflammation, attenuation, ulceration, and thickness) at 21 days of posthatch age (**doa**).

| | | Non-Injected [1] | Saline [2] | L-AA 12 [3] | L-AA 25 [4] | Pooled SEM | *p*-Value |
|---|---|---|---|---|---|---|---|
| | | | | **Treatments** | | | |
| Cilial Erosion [5] (%) | Score 0 | 50.0 | 50.0 | 75.0 | 75.0 | 38.19 | 0.835 |
| | Score 1 | 27.3 | 8.3 | 8.3 | 8.3 | 14.14 | 0.460 |
| | Score 2 | 18.2 | 25.0 | 25.0 | 33.3 | 18.89 | 0.884 |
| | Score 3 | 0 | 0 | 0 | 0 | - | - |
| Glands [6] (%) | Score 0 | 18.2 | 8.3 | 0.0 | 16.7 | 13.08 | 0.478 |
| | Score 1 | 72.7 | 91.7 | 83.3 | 66.7 | 17.35 | 0.475 |
| | Score 2 | 0 | 0 | 0 | 0 | - | - |
| | Score 3 | 9.1 | 0 | 16.7 | 16.7 | 13.11 | 0.521 |
| Inflammation [7] (%) | Score 0 | 18.2 | 8.3 | 0 | 16.7 | 13.08 | 0.478 |
| | Score 1 | 63.6 | 91.7 | 83.3 | 66.7 | 17.77 | 0.337 |
| | Score 2 | 0 | 0 | 0 | 0 | - | - |
| | Score 3 | 27.3 | 0 | 16.7 | 16.7 | 14.94 | 0.340 |

**Table 7.** *Cont.*

| | | Non-Injected [1] | Saline [2] | L-AA 12 [3] | L-AA 25 [4] | Pooled SEM | *p*-Value |
|---|---|---|---|---|---|---|---|
| | | | **Treatments** | | | | |
| Attenuation [8] (%) | Score 0 | 27.3 | 16.7 | 0 | 16.7 | 14.95 | 0.340 |
| | Score 1 | 36.4 [a] | 8.3 [b] | 0 [b] | 8.3 [b] | 13.32 | **0.017** |
| | Score 2 | 27.3 [b] | 66.7 [a] | 91.6 [a] | 66.6 [a] | 18.48 | **0.006** |
| | Score 3 | 9.1 | 0 | 8.3 | 8.3 | 10.54 | 0.793 |
| Ulceration [9] (%) | Score 0 | 54.6 | 83.3 | 83.3 | 83.3 | 17.70 | 0.288 |
| | Score 1 | 36.4 | 8.3 | 16.7 | 8.3 | 15.65 | 0.254 |
| | Score 2 | 0 | 8.3 | 0 | 8.30 | 8.62 | 0.595 |
| | Score 3 | 3.9 | 0 | 0 | 0 | 2.63 | 0.360 |
| Thickness [10] (μm) | | 37.3 | 39.8 | 35.4 | 40.5 | 5.60 | 0.778 |

[a,b] Treatment means within the same variable column within type of treatment with no common superscript differ significantly ($p \leq 0.05$). [1] Eggs that were not injected with solution. [2] Eggs that were injected with 100 μL saline at 17 doi. [3] Eggs that were injected with 100 μL saline containing L-AA 12 at 17 doi. [4] Eggs that were injected with 100 μL saline containing L-AA 25 at 17 doi. [5] Cilial erosions were scored as follows: 0 = (normal), 1 = focal (mild), 2 = multifocal (moderate), 3 = diffuse (severe). [6] Tracheal glands were scored as follows: 0 = normal, 1 = depletion (mild), 2 = hyperplasia (moderate), 3 = severe. [7] Tracheal inflammation was scored as follows: 0 = normal, 1 = mild lymphocytes, 2 = moderate lymphocytes, 3 = heterophils/moderate lymphocytes. [8] Attenuations were scored as follows: 0 = normal, 1 = focal (mild), 2 = multifocal (moderate), 3 = diffuse (severe). [9] Ulcerations were scored as follows: 0 = normal, 1 = focal (mild), 2 = multifocal (moderate), 3 = diffuse (severe). [10] Tracheal thickness (μm) was recorded as an average of three measurements in different locations within each sample. N = One bird in each of the 12 replicate groups in each treatment–score combination was used for means calculations.

**Table 8.** Effects of treatment (non-injected; saline-injected (**saline**); saline containing 12 mg of L-ascorbic acid (**L-AA 12**) or 25 mg of L-ascorbic acid (**L-AA 25**) administered at 17 days of incubation (**doi**)) on percentages of 0, 1, 2, and 3 scores for various tracheal histomorphological variables (cilial erosion, glands, inflammation, attenuation, ulceration, and thickness) at 28 days of posthatch age (**doa**).

| | | Non-Injected [1] | Saline [2] | L-AA 12 [3] | L-AA 25 [4] | Pooled SEM | *p*-Value |
|---|---|---|---|---|---|---|---|
| | | | **Treatments** | | | | |
| Cilial Erosion [5] (%) | Score 0 | 25.0 | 16.7 | 58.3 | 41.7 | 19.22 | 0.152 |
| | Score 1 | 33.3 | 33.3 | 25.0 | 25.0 | 19.30 | 0.945 |
| | Score 2 | 33.3 | 50.0 | 16.7 | 25.0 | 19.05 | 0.353 |
| | Score 3 | 8.3 | 0 | 0 | 8.3 | 8.33 | 0.577 |
| Glands [6] (%) | Score 0 | 0 [b] | 8.3 [b] | 66.7 [a] | 41.7 [ab] | 18.21 | **0.018** |
| | Score 1 | 33.3 | 33.3 | 16.7 | 25.0 | 18.72 | 0.780 |
| | Score 2 | 33.3 | 58.3 | 16.7 | 25.0 | 18.97 | 0.161 |
| | Score 3 | 8.3 | 0 | 0 | 8.3 | 8.33 | 0.577 |
| Inflammation [7] (%) | Score 0 | 8.3 | 8.3 | 25.0 | 8.3 | 13.76 | 0.538 |
| | Score 1 | 91.7 [a] | 91.7 [a] | 50.0 [b] | 83.3 [a] | 15.69 | **0.033** |
| | Score 2 | 0 | 0 | 16.7 | 8.3 | 9.89 | 0.286 |
| | Score 3 | 0 | 0 | 8.3 | 0 | 5.89 | 0.402 |
| Attenuation [8] (%) | Score 0 | 8.3 | 8.3 | 33.3 | 25.0 | 15.99 | 0.313 |
| | Score 1 | 8.3 | 0.0 | 16.7 | 33.3 | 14.10 | 0.123 |
| | Score 2 | 16.7 | 66.7 | 41.7 | 33.3 | 19.38 | 0.089 |
| | Score 3 | 66.7 [a] | 25.0 [b] | 8.3 [b] | 8.3 [b] | 15.99 | **0.002** |

**Table 8.** *Cont.*

| | | Treatments | | | | Pooled SEM | *p*-Value |
|---|---|---|---|---|---|---|---|
| | | **Non-Injected** [1] | **Saline** [2] | **L-AA 12** [3] | **L-AA 25** [4] | | |
| Ulceration [9] (%) | Score 0 | 100.0 | 100.0 | 100.0 | 83.3 | 7.95 | 0.102 |
| | Score 1 | 0 | 0 | 0 | 8.3 | 5.89 | 0.402 |
| | Score 2 | 0 | 0 | 0 | 8.3 | 5.89 | 0.402 |
| | Score 3 | 17.3 | 14.3 | 15.6 | 17.4 | 2.24 | 0.446 |
| Thickness [10] (μm) | | 25.8 | 24.4 | 25.7 | 20.9 | 2.44 | 0.186 |

[a,b] Treatment means within the same variable column within type of treatment with no common superscript differ significantly ($p \leq 0.05$). [1] Eggs that were not injected with solution. [2] Eggs that were injected with 100 μL saline at 17 doi. [3] Eggs that were injected with 100 μL saline containing L-AA 12 at 17 doi. [4] Eggs that were injected with 100 μL saline containing L-AA 25 at 17 doi. [5] Cilial erosions were scored as follows: 0 = (normal), 1 = focal (mild), 2 = multifocal (moderate), 3 = diffuse (severe). [6] Tracheal glands were scored as follows: 0 = normal, 1 = depletion (mild), 2 = hyperplasia (moderate), 3 = severe. [7] Tracheal inflammation was scored as follows: 0 = normal, 1 = mild lymphocytes, 2 = moderate lymphocytes, 3 = heterophils/moderate lymphocytes. [8] Attenuations were scored as follows: 0 = normal, 1 = focal (mild), 2 = multifocal (moderate), 3 = diffuse (severe). [9] Ulcerations were scored as follows: 0 = normal, 1 = focal (mild), 2 = multifocal (moderate), 3 = diffuse (severe). [10] Tracheal thickness (μm) was recorded as an average of three measurements in different locations within each sample. N = One bird in each of the 12 replicate groups in each treatment–score combination was used for means calculations.

## 4. Discussion

The aim of the current study was to determine the effects of the *in ovo* administration of various amounts of L-AA on eye and liver L-AA concentrations, plasma NO and mineral concentrations, and tracheal histomorphology in broilers exposed to 50 ppm of atmospheric $NH_3$ from 0 to 28 doa. The effects of feed additives, including various enzymes, minerals, and vitamins, on trachea histomorphology, eye and liver L-AA concentrations, and plasma NO concentrations in chickens and livestock subjected to elevated atmospheric $NH_3$ levels have not been previously investigated. It was hypothesized that the resistance of developing broilers to the negative physiological impacts caused by high atmospheric $NH_3$ levels in the posthatch period would be improved by an earlier exposure to supplemental L-AA through *in ovo* injection. However, no significant treatment differences were observed for plasma NO, Ca, Cu, or Zn concentrations, or eye and liver L-AA concentrations of the birds at any of the designated posthatch time periods while they were exposed to elevated atmospheric levels of $NH_3$. Because the *in ovo* L-AA treatments employed had no effect on the liver or eye L-AA concentrations of the birds exposed to elevated aerial $NH_3$ levels, it is suggested that any noted effects of treatment were not mediated by increased concentrations of L-AA in those tissues.

Nitric oxide is of paramount importance as a mediator of inflammatory responses and is produced by the oxidation of L-arginine by NO synthase. The inducible NO synthase enzyme is important for immunity and inflammation [62]. The expression of cytokines, such as IL-1β, IL-12, interferon-γ, and tumor necrosis factor, increase the production of NO [63], and the production of NO, as well as other inflammatory cytokines (TNF-α, IL-6, IL-1β, and TLR-2A), has been shown to increase in the bursa of Fabricius of broilers exposed long-term to atmospheric $NH_3$ levels [64]. Cytokines play an important role in the course of an inflammatory response [65], and the overproduction of molecules, such as IL-1β and IL-6, which are potent proinflammatory and immunomodulatory cytokines [66], could have potentially harmful effects on feed efficiency and growth in different species [67,68]. These findings also demonstrated that the production of NO and other inflammatory cytokines increases in sites of infection, such as the bursa of Fabricius. Nevertheless, in the current study, no changes in plasma NO concentration were observed at any of the time periods investigated. This would indicate that no systemic inflammation occurred in response to the subjection of the birds to 50 ppm of atmospheric $NH_3$. Further study is needed to determine whether NO production as well as other inflammatory indicators occur in

the trachea, where some levels of infection were observed in broilers subjected to chronic elevations of aerial $NH_3$.

Although systemic inflammation was apparently not significantly affected by treatment, the trachea histomorphology results revealed that the *in ovo* injection of 12 mg of L-AA appeared to be associated with reduced tracheal inflammation at 28 doa compared to the histomorphology of birds receiving control treatments. Therefore, the current results suggest that the *in ovo* injection of 12 mg of L-AA may attenuate an inflammatory response after chronic exposure to high atmospheric $NH_3$ levels. The submucosa of the tracheal wall contains mucous glands, which secrete mucus to facilitate the entrapment of particles, and cilia (tiny hairlike structures present on the surface of tracheal epithelial cells), which propel entrapped particles for disposal. Tracheal cilia, therefore, assist in the removal of potentially harmful fluids and particles from the airways and lungs [58,59]. When erosion occurs, the ciliary function is impaired, and epithelial hypoplasia occurs [69]. In affected areas, a process known as attenuation causes ciliated columnar epithelial cells to appear abnormally shortened in height with multiple defects, including ulceration [59]. Birds in the L-AA 12 and 25 *in ovo* injection treatment groups exhibited less epithelial cell attenuation at 0 doa compared to birds in the non-injected and saline-injected control groups and at 21 doa compared to those in the non-injected control group. Birds that received the L-AA 12 or 25 treatments also had fewer sites of ulceration in the trachea and fewer areas of open mucous membranes at 0 doa compared to the non-injected control treatment. A reduction in inflammation incidence and the presence of tracheal lymphocytes were also observed in birds belonging to the L-AA 12 treatment group alone. Therefore, the L-AA *in ovo* injection treatment promoted the establishment of these ciliated epithelial cells and could thereby improve the tracheal function.

In the immune system, the major role of L-AA as a physiological antioxidant is to protect host cells against oxidative stress caused by infections [70]. It is well documented that $NH_3$ exposure is highly associated with systematic and local oxidative stress [71]. The exposure of broilers to atmospheric $NH_3$ has been shown to result in an increase in the concentrations of free radical indicators, including gamma-glutamyl transferase, malondialdehyde (MDA), and hydrogen peroxide, and a decrease in the antioxidant enzymatic activities of catalase, glutathione, and glutathione peroxidase in the bursa of Fabricius [71]. In addition, aerial $NH_3$ is linked to impaired immune function, where the numbers of CD8+ T-lymphocytes and the activity of adenosine triphosphate are reduced in broilers subjected long-term to elevated atmospheric $NH_3$ concentrations [72]. Furthermore, an increase in antioxidant activity is linked to a reduction in systematic and local oxidative stress [73,74]. Therefore, an increase in antioxidant activity leading to an improvement in local immunity may be beneficial during chronic exposure to elevated atmospheric $NH_3$ levels.

In our laboratory, the *in ovo* injection of 12 and 25 mg of L-AA was previously found to improve the enzymatic (serum superoxide dismutase concentrations) and non-enzymatic (MDA) activities in broilers under normal conditions [8]. In the companion study, broilers that received 12 mg of *in ovo* injected L-AA exhibited lower incidence of corneal erosion and higher gain of body weight between 0 and 28 doa compared to other treatment groups while being chronically exposed to elevated atmospheric $NH_3$ [49]. Therefore, improved ocular and trachea histomorphological features could be partially linked to increased systemic and local antioxidant activities in broilers chronically exposed to elevated atmospheric $NH_3$. However, further research is needed to determine the effects of various levels of *in ovo* injected L-AA on the enzymatic and non-enzymatic antioxidant capacities of birds in response to elevated aerial $NH_3$ concentrations.

Previous studies have reported a decrease in the concentrations of plasma trace minerals, such as sodium (Na+) and chloride (Cl-), in broilers subjected to 50 ppm of aerial $NH_3$ [47]. However, dietary supplementation of L-AA at 250 mg/kg has been reported to have no significant effect on the plasma levels of Ca, Cu, and Zn in broilers experiencing oxidative stress induced by Cu toxicity [75]. Furthermore, the plasma levels of Cu have not been determined in chickens provided L-AA supplementation in the diet or by *in ovo*

injection while housed under commercial conditions. In addition to these previous reports, the plasma minerals measured in this study were not affected by the levels of L-AA that were administered by *in ovo* injection while the birds were being continually exposed to elevated atmospheric $NH_3$ levels. Hence, the treatment effects noted for the broilers in this study did not involve changes in their plasma levels of Ca, Cu, or Zn.

## 5. Conclusions

In conclusion, the results of this study showed that the *in ovo* injection of 12 or 25 mg of L-AA did not significantly affect the plasma NO and mineral concentrations, as well as the tissue L-AA concentrations, of the broilers between 0 and 28 doa. However, the *in ovo* injection of 12 mg of L-AA was associated with decreased inflammation and improved histologic changes of the trachea of the birds while they were exposed to elevated atmospheric levels of $NH_3$. The promising results concerning tracheal histomorphology could be linked to the antioxidant and immunomodulatory activities of L-AA, which have been observed in previous studies. Because the beneficial results reported in earlier studies were observed in birds housed under normal conditions, further studies should continue to explore the potential modulatory influences of other levels of supplemental L-AA administered by *in ovo* injection on tracheal histomorphology, local antioxidant and immunomodulatory activities, and the systemic and local inflammatory responses of broilers chronically subjected to high levels of aerial $NH_3$.

**Author Contributions:** Conceptualization, A.M., S.A.F. and E.D.P.; methodology, A.M., S.A.F., J.L.P., H.A.O., B.B., K.M. and E.D.P.; software, S.A.F.; validation, A.M., S.A.F., B.B., K.M., A.W.L. and E.D.P.; formal analysis, P.D.G., A.M. and S.A.F.; investigation, A.M.; resources, E.D.P.; data curation, A.M., S.A.F. and E.D.P.; writing—original draft preparation, A.M.; writing—review and editing, A.M., S.A.F., B.B., K.M., H.A.O. and E.D.P.; visualization, A.M., S.A.F. and E.D.P.; supervision, S.A.F. and E.D.P.; project administration, A.M.; funding acquisition, E.D.P. All authors have read and agreed to the published version of the manuscript.

**Funding:** This research was supported by the United States Department of Agriculture (USDA agreement no. 58-6064-9-016), DSM Nutritional Products Inc., Merial Select Inc., and Zoetis Animal Health Co.

**Institutional Review Board Statement:** The experimental procedure was approved by the Mississippi State University Institutional Animal Care and Use Committee (Protocol #IACUC-20-248).

**Informed Consent Statement:** Not applicable.

**Data Availability Statement:** None of the data were deposited in an official repository.

**Acknowledgments:** The authors express their appreciation for the assistance provided by the graduate and undergraduate students in the Poultry Science Department at Mississippi State University and for the invaluable support of Robert Wills.

**Conflicts of Interest:** The authors indicate that there are no conflict of interest.

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
