# Peer review of "Effects of the In ovo Administration of L-Ascorbic Acid on Tissue L-Ascorbic Acid Concentrations, Systemic Inflammation, and Tracheal Histomorphology of Ross 708 Broilers Subjected to Elevated Levels of Atmospheric Ammoniaâ€"

_poultry, doi:10.3390/poultry2020014_

Round 1
Reviewer 1 Report
Dear Authors,
You have made 10 references to yourself in your study. Academically, it may not be appropriate to cite so much yourself. Some references seem to be made unnecessarily. These need to be checked and corrected.
Line 56: It is not appropriate to include reference number 4 (Peebles, 2018) here. Because this is a compilation, not a research manuscript.
Line 114: Please replace “fertile hatching eggs” with "broiler hatching eggs". It's possible that the initial fertilization of the eggs is unknown.
Lines 114-116: You said that “stored for 24 hours prior to set in accordance with the procedure described by Fatemi et al [12].”. There is no information about the storage conditions you mentioned in this article. Please delete this reference or add the other reference.
Lines 120-122: You said that “Air temperature and relative humidity were monitored according to the method described by Fatemi et al [6,11].” Please delete this references. Controlling the temperature and humidity in the machine is not a method. It is not appropriate to give reference in this way.
Lines 162-163: The liver and eye L-AA concentrations were measured according to the method described by Mousstaaid et al [7,8]. Please delete this reference.
Lines 172-173: Plasma NO concentrations were determined using the method described by Fatemi et al. [17]. Please delete this reference.
Lines 124-139: In addition to the number of eggs you specified at the beginning of incubation, indicate how many eggs were injected on the 17th day of incubation and how many eggs were in each group and subgroup.
Check the superscripts in Table 3.
Lines 268 and 360: Replaced “35 doa” with “28 doa”. In the method part, it was stated that 28-day characteristics were determined.
Author Response
Reviewer 1:
- You have made 10 references to yourself in your study. Academically, it may not be appropriate to cite so much yourself. Some references seem to be made unnecessarily. These need to be checked and corrected.
Answer:
The numbers of the aforementioned references have been reduced. Furthermore, we corrected the manuscript for similarities which are mostly linked to methodologies used from these references. The areas that have been cited are mostly introduced by only those specific references utilized and for which another reference cannot be used as a replacement. However, we tried to keep those references as low as possible. The deleted references are listed below
- Fatemi, S.A.; Elliott, K.E.C.; Bello, A.; Zhang, H.; Peebles, E.D. Effects of the in ovo injection of vitamin D3 and 25-hydroxyvitamin D3 in Ross 708 broilers subsequently fed commercial or calcium and phosphorous-restricted diets: II. Immunity and small intestine morphology. Sci. 2021, 100, 101240. https://doi.org/10.1016/j.psj.2021.101240.
- Fatemi, S.A.; Elliott, K.E.C.; Bello, A.; Macklin, K.S.; Peebles, E.D. Effects of the in ovo injection of vitamin D3 and 25-hydroxyvitamin D3 in Ross 708 broilers subsequently challenged with coccidiosis: II. Immunological and inflammatory responses and small intestine histomorphology. Animals Basel 2022, 12, 1027. https://doi.org/10.3390/ani12081027.
- Peebles, E.D. In ovo applications in poultry: A review. Poult. Sci. 2018, 97, 2322–2338.
- Line 56: It is not appropriate to include reference number 4 (Peebles, 2018) here. Because this is a compilation, not a research manuscript.
Answer:
The reference number 4 was removed
- Line 114: Please replace “fertile hatching eggs” with "broiler hatching eggs". It's possible that the initial fertilization of the eggs is unknown.
Answer:
The relevant correction was applied to this section
- Lines 114-116: You said that “stored for 24 hours prior to set in accordance with the procedure described by Fatemi et al [12].”. There is no information about the storage conditions you mentioned in this article. Please delete this reference or add the other reference.
Answer:
The relevant correction was applied to this section
- Lines 120-122: You said that “Air temperature and relative humidity were monitored according to the method described by Fatemi et al [6,11].” Please delete this references. Controlling the temperature and humidity in the machine is not a method. It is not appropriate to give reference in this way.
Answer:
Air temperature and relative humility are the main two important factors that affect embryo growth and it is very crucial to ensure all eggs incubated in the similar condition in order to eliminate the any possible confounding factor that influence hatch process and hatchling quality. Therefore, the opportunity to monitor the air temperature and relative humidity are considered as the strength of current study and authors prefer to keep this section with relevant references.
- Lines 162-163: The liver and eye L-AA concentrations were measured according to the method described by Mousstaaid et al [7,8]. Please delete this reference.
Answer:
Mousstaaid et al [7,8] are the only references that explained in detail the procedure of measuring L-AA in serum as well as eye in chicken and we exactly followed the same procedure for measuring the L-AA concentration in the liver and eye. Therefore, the proper references for this section are Mousstaaid et al [7,8] which do not allow us to remove them.
- Lines 172-173: Plasma NO concentrations were determined using the method described by Fatemi et al. [17]. Please delete this reference.
Answer:
The relevant correction was applied to this section.
- Lines 124-139: In addition to the number of eggs you specified at the beginning of incubation, indicate how many eggs were injected on the 17thday of incubation and how many eggs were in each group and subgroup.
Answer:
The relevant corrections were applied on lines 131-133.
“Approximately 350 eggs per each pre-specified treatment were injected using a Zoetis Inovoject® m (Zoetis Animal Health, Research Triangle Park, NC, USA) multi-egg injection machine.”
- Check the superscripts in Table 3.
Answer:
The relevant corrections were applied to Table 3.
- Lines 268 and 360: Replaced “35 doa” with “28 doa”. In the method part, it was stated that 28-day characteristics were determined.
Answer:
The relevant corrections were applied on lines 268 and 360.

Reviewer 2 Report
The manuscript was written at bad logical, all the manuscript was confusion, including Introduction, Materials and Methods, and results even if being revision.The in ovo injection can improve development of broiler just at early phase.However, this research stuied the broilers until 28 d of age.
This research measure L-AA concentration in eye.why choose eye to measure the L-AA concentration isn't reviewed in introduction.
Author Response
Reviewer 2:
- The manuscript was written at bad logical, all the manuscript was confusion, including Introduction, Materials and Methods, and results even if being revision.
The reviewer does not specifically describe what portions of the section mentioned need revising. Therefore, all the authors are not clear as to what specific information in the paper is confusing, so that appropriate responses could be provided. The type of content and structure of the description in those sections conform to previous publications including previous publications in the Poultry journal. Also, this is contrary to general opinion of reviewer #1.
- The in ovo injection can improve development of broiler just at early phase. However, this research stuied the broilers until 28 d of age.
There are several studies related to many in ovo-injected materials, including vitamins, minerals, carbohydrates, antioxidants, hormones, and vaccines that exhibits the longer term effects in not only broilers, but also laying pullets. We also listed many relevant studies in the Introduction and Discussion sections that showed the effects of various doses of in ovo injection of L-AA at processing age (42 day) which later than 28 day. Furthermore, we observed the later effects of in ovo injection of L-AA from our pervious published studies on live performance, antioxidant activity and meat quality at the processing age when the same treatments in the current study were tested.
- This research measure L-AA concentration in eye. why choose eye to measure the L-AA concentration isn't reviewed in introduction.
The relevant corrections were applied to the introduction on lines 96-102.
“The long term exposure to NH3 has been shown to result in sever ocular abnormalities and infection due to an increase oxidative reaction and inflammation [44]. Other associated physiological effects include severe ocular abnormalities [44] and irritation of mucous membranes leading to corneal ulceration and tracheitis [41,45]. Shi et al. [46] also showed that exposing broilers for 6 weeks to aerial NH3 concentrations between 19.5 and 45.5 ppm caused abnormalities in their tracheal immune response. L-ascorbic acid is considered antioxidant agents and it is highly involved in reduction of systemic inflammation. In addition, our laboratory found that male broilers have higher L-AA concentration as compared to female at hatch [7].”
